# Electron Tomography as a Tool to Study SARS-CoV-2 Morphology

**DOI:** 10.3390/ijms252111762

**Published:** 2024-11-01

**Authors:** Hong Wu, Yoshihiko Fujioka, Shoichi Sakaguchi, Youichi Suzuki, Takashi Nakano

**Affiliations:** Department of Microbiology and Infection Control, Faculty of Medicine, Osaka Medical and Pharmaceutical University, Osaka 565-0871, Japan; yoshihiko.fujioka@ompu.ac.jp (Y.F.); shoichi.sakaguchi@ompu.ac.jp (S.S.); tnakano@ompu.ac.jp (T.N.)

**Keywords:** SARS-CoV-2, electron microscopy, transmission electron microscopy, electron tomography, three-dimensional reconstruction, morphogenesis, virus budding

## Abstract

Severe acute respiratory syndrome coronavirus 2 (SARS-CoV-2), a novel betacoronavirus, is the causative agent of COVID-19, which has caused economic and social disruption worldwide. To date, many drugs and vaccines have been developed for the treatment and prevention of COVID-19 and have effectively controlled the global epidemic of SARS-CoV-2. However, SARS-CoV-2 is highly mutable, leading to the emergence of new variants that may counteract current therapeutic measures. Electron microscopy (EM) is a valuable technique for obtaining ultrastructural information about the intracellular process of virus replication. In particular, EM allows us to visualize the morphological and subcellular changes during virion formation, which would provide a promising avenue for the development of antiviral agents effective against new SARS-CoV-2 variants. In this review, we present our recent findings using transmission electron microscopy (TEM) combined with electron tomography (ET) to reveal the morphologically distinct types of SARS-CoV-2 particles, demonstrating that TEM and ET are valuable tools for visually understanding the maturation status of SARS-CoV-2 in infected cells. This review also discusses the application of EM analysis to the evaluation of genetically engineered RNA viruses.

## 1. Introduction

Severe acute respiratory syndrome coronavirus 2 (SARS-CoV-2) is the causative agent of COVID-19, which is associated with severe symptoms, including acute respiratory distress syndrome, vasculitis, thrombosis, stroke, myocardial damage, and multiple organ failure [1,2]. Since the first case was confirmed in Wuhan, China, in late 2019, it has been confirmed that over 700 million people have been infected with, and over 7 million patients have died from, SARS-CoV-2 worldwide [3]. Currently, vaccines to prevent SARS-CoV-2 infection and therapeutic agents, including small-molecule drugs and neutralizing monoclonal antibodies against SARS-CoV-2, are in practical use and are effective in stopping the spread of COVID-19. In May 2023, WHO declared that COVID-19 was no longer a global emergency [4]. However, even though COVID-19 could be considered an epidemic infectious disease like seasonal influenza, SARS-CoV-2 is highly mutable, leading to the emergence of new variants that could counteract the efficacy of existing prophylactic or therapeutic measures [5]. Therefore, COVID-19 remains a health threat, and it is important to understand the biological characteristics of SARS-CoV-2 and apply this knowledge to the development of new preventive and therapeutic interventions to control COVID-19.

SARS-CoV-2 is an enveloped virus with a positive single-stranded RNA genome of approximately 26–32 kb and belongs to the genus Betacoronavirus of the Coronaviridae family, which includes other medically important coronaviruses responsible for severe acute respiratory syndrome (SARS) and Middle East respiratory syndrome (MERS) [5,6,7,8]. The SARS-CoV-2 virion consists of the spike (S), envelope (E), membrane (M), and nucleocapsid (N) proteins, as well as viral RNA. Two-thirds of the viral genomic RNA is occupied by two open reading frames (ORFs), termed ORF1a and ORF1b, which encode 16 non-structural proteins (NSPs) that function in the replication–transcription complex (RTC), while structural (i.e., S, E, M, and N) proteins are expressed from the ORFs located in the 3′ third of the genome (Figure 1A) [8].

Replication of SARS-CoV-2 begins with the attachment of virus particles to the surface receptors of target cells. Angiotensin-converting enzyme 2 (ACE2) has been shown to be a functional receptor for SARS-CoV-2, and the S protein located on the outermost part of the virus particle is responsible for binding to ACE2 [12,13]. Importantly, mutations in the S protein have a significant effect not only on infectivity, but also on the susceptibility of the virus to neutralizing antibodies induced by vaccination. Therefore, adaptive mutations in the S protein alter the transmission and pathogenicity of SARS-CoV-2, complicating the development of COVID-19 prevention and treatment methods [5]. The S protein is a fusion glycoprotein composed of S1 and S2 subunits [13]. Although receptor binding is mediated by the receptor-binding domain (RBD) of the S1 subunit, upon cleavage of the S1/S2 junction by cellular proteases, including transmembrane protease serine 2 (TMPRSS2), the exposed S2 subunit induces fusion between viral and cellular membranes [12,13,14]. Internalization of the SARS-CoV-2 virion occurs via endocytosis. Initial translation of the internalized viral RNA occurs in the cytoplasm, producing two large polyproteins from ORF1a and ORF1b, and viral RNA amplification occurs on a unique organelle-like structure, the viral replication organelle, formed in the endoplasmic reticulum (ER) [15,16,17]. Subsequently, translated structural proteins translocate to the lumen of the ER-to-Golgi intermediate compartment (ERGIC) and are assembled with viral RNA, resulting in the budding of immature virus particles into the lumen of the ERGIC [16,18]. In the final stage of SARS-CoV-2 replication, virus particles undergo post-translational modifications of structural proteins via trafficking to the Golgi and trans-Golgi network and egress via lysosomal exocytosis (Figure 1B) [19].

To prevent the spread of SARS-CoV-2, diagnosis is quite important in clinical practice, and therefore, many molecular and serological diagnostic methods to detect the viral components have been developed and deployed (comprehensively reviewed in [20]). As a means of detecting whole viruses within cells and tissues, electron microscopy (EM) is a straightforward method that allows for direct visualization of viruses in specimens. In the current clinical setting, EM is not the first-line test method to prove the presence of viruses in specimens, particularly for acute infectious diseases that require rapid diagnosis. However, when a clinician or pathologist wants to know if a disease is caused by a virus, EM plays a key role in identifying the causative agent of the disease, as demonstrated in earlier reports of SARS and COVID-19 [21,22,23,24]. In basic research, EM remains a powerful technique for studying the morphological features of the virion and the biological features of virus replication. The recent development of cryogenic electron microscopy (cryo-EM) has allowed for the ultrastructural analysis of viruses in their native state [25], and cryo-EM has been used to determine the molecular architecture and organization of SARS-CoV-2 [26,27,28]. In addition, the application of cryo-electron tomography (cryo-ET), which is the tomographic reconstitution of cryo-EM micrographs, allowed for three-dimensional (3D) in situ imaging of intracellular events in SARS-CoV-2 replication [16,17,29].

Compared to cryo-EM, transmission electron microscopy (TEM) is a conventional ultrastructural imaging technique. Although many groundbreaking morphological studies using TEM were actively carried out in the 1950s and 1960s, the rapid development of simpler technologies in biochemistry and molecular biology since the 1970s, which can detect the chemical properties of viruses, has diminished the importance of TEM in basic research. Nevertheless, in the field of virology, it is always a natural desire of the researcher to see the virus itself. Notably, it is a simpler and less expensive way to analyze the exact localization of viruses in tissues and within cells. In addition, TEM can be used to study interactions between virus particles and organs. The resolution of TEM is typically in the nanometer range, making it suitable for large-scale structural analysis of cells and tissues. Although it is difficult to achieve the atomic resolution that can be reached by cryo-EM, TEM is excellent for observing extensive structures within a sample. Thus, TEM remains a powerful tool in the molecular morphology of SARS-CoV-2. For example, pathologists often use the TEM technique to rapidly detect SARS-CoV-2 in the tissue samples of COVID-19 patients [23]. In this review, we focus our attention on the usefulness of conventional TEM in the analysis of the viral ultrastructure. In addition, we have shown that TEM can also be applied to 3D morphological analysis of SARS-CoV-2 particles (electron tomography [ET], Figure 2) and revealed the presence of different types of virus particles in infected cells, which would indicate the different maturation states of virus particles within cells [30,31,32]. Therefore, TEM and its 3D application to ET still serve as a complementary method for in situ analysis of the behavior of SARS-CoV-2 [32].

## 2. Current Applications of TEM and ET in Viral Analysis

Cryo-EM is ideal for detailed structural studies at high resolution, particularly for macromolecular complexes and viruses. In contrast, TEM is more accessible and is widely used for broader structural analysis within complex biological systems (Figure 3). Cryo-EM allows for imaging of biological samples in their near-native, frozen-hydrated state without chemical fixation or staining [33]. In cryo-EM, samples are rapidly frozen, preserving their native structure, and then imaged to obtain high-resolution two-dimensional (2D) projections. These projections can be computationally averaged to produce detailed 3D reconstructions, which are particularly useful for studying macromolecular complexes. Cryo-ET, a specialized application of cryo-EM, involves tilting the sample to capture images from multiple angles, which are then reconstructed into a 3D structure. This method is particularly advantageous for studying the architecture of viruses and cellular structures in a state as close as possible to their natural environment. On the other hand, TEM typically requires the chemical fixation, dehydration, and sectioning of samples, which can introduce artifacts but allow for the observation of larger and more complex biological systems [34]. TEM provides high-resolution 2D images, enabling the detailed observation of fine structural features. As mentioned above, TEM can be combined with ET, which generates 3D reconstructions of a specific area by tilting the sample and capturing multiple images, similar to cryo-ET but generally at a broader scale (Figure 3) [35]. It is important to note that cryo-EM typically averages data from multiple identical particles to enhance resolution, while TEM (and ET) often focuses on the detailed study of individual particles or sections [36]. Thus, while both techniques are invaluable in structural biology, they have distinct characteristics and should be chosen based on the specific requirements of the research.

Applying both TEM and cryo-EM in the study of SARS-CoV-2 has provided complementary insights into the virus’s structural and functional characteristics, aiding in developing therapeutic strategies and vaccines. TEM studies have revealed that SARS-CoV-2 virions are typically spherical or pleomorphic, with a diameter ranging from 60 to 140 nm [37]. These virions possess an envelope derived from the host cell membrane and studded with S proteins that give the virus its characteristic crown-like appearance [38]. The internal structure, as visualized by TEM, includes a helical nucleocapsid that encapsulates the viral RNA genome, and the virus replicates and assembles within the cytoplasm of infected cells, maturing on the inner membrane of the rough ER [18,39]. TEM and tomogram observed SARS-CoV-induced membrane changes [40]. TEM also demonstrated the exocytosis of virus particles and their aggregation on the cell membrane surface, as well as ultrastructural changes in infected cells indicative of apoptosis, particularly in type II alveolar epithelial cells observed in bronchoalveolar lavage fluid [41,42]. Cryo-ET, a more advanced variant of cryo-EM, has provided high-resolution 3D views of SARS-CoV-2, elucidating detailed structural characteristics that are not as easily observed with traditional TEM. Cryo-ET has confirmed the roughly spherical nature of the virus, with an average diameter of approximately 89.8 nm, and has highlighted the mixed conformational states of the spike proteins on the virion surface, which are crucial for host cell entry and fusion [16]. Additionally, cryo-ET has offered insights into the internal organization of the virus, revealing how the virus packs its approximately 30 kb-long RNA genome into its lumen [26,43]. The replication and assembly processes observed via cryo-ET include the formation of double-membrane vesicles (DMVs) within infected cells, serving as replication organelles, and the assembly of virions at the ERGIC [29]. Cryo-ET also sheds light on the structural variations among SARS-CoV-2 variants, such as the Omicron lineages BA.1 and BA.2, which display distinct spike protein characteristics that may influence transmissibility and immune evasion. In addition to the contributions of cryo-ET, ET has also proven valuable in virology. One advantage of ET is its ability to operate at room temperature, allowing for the observation of biological samples without cryogenic conditions. This feature simplifies sample preparation and makes the technique more accessible. Traditional ET is particularly beneficial for analyzing thicker samples and providing high-resolution images of viral structures within cellular contexts. In the case of SARS-CoV-2, 3D reconstructions achieved through traditional ET have revealed the ultrastructural features of virus particles within infected cells [30]. This approach has been instrumental in understanding the morphology, spatial distribution, and components of virions, complementing cryo-ET findings and broadening the scope of virological research. Traditional ET has also been used effectively to study the structural dynamics of viral assembly and maturation processes, providing a broader understanding of these critical steps in the viral life cycle. Together, TEM, cryo-ET, and ET offer a comprehensive view of SARS-CoV-2, from its overall morphology and replication mechanisms to intricate structural details at the molecular level, thus playing a critical role in the ongoing battle against COVID-19.

The application of both TEM and cryo-ET in the study of human immunodeficiency virus (HIV) has provided deep insights into the virus’s structure, assembly, maturation, and entry mechanisms, which are crucial for developing effective therapeutic strategies and vaccines against HIV/AIDS. TEM studies have revealed that HIV particles are spherical in shape, possessing an envelope derived from the host cell membrane [44]. The virus contains a distinctive cone-shaped core within this envelope, encapsulating the viral RNA genome. TEM has also confirmed that mature HIV-1 particles contain two copies of RNA strands within the core, forming an interwound, coiling structure [45]. These RNA strands are already present during the late budding stage, suggesting their incorporation into the virion during assembly. An advanced TEM technique called localization microscopy (iPALM) has enabled detailed visualization of internal viral structures, including the capsid, revealing the pleomorphic nature of HIV capsids [46]. Furthermore, using inositol hexakisphosphate (IP6) to stabilize the capsid structure has prevented premature disassembly during sample preparation, highlighting methodological advancements that have significantly contributed to understanding the virus’s life cycle [47,48]. In contrast, cryo-ET has provided high-resolution 3D views of HIV in its native state, offering detailed insights into various aspects of the virus. Cryo-ET has been used to visualize the roughly spherical structure of HIV virions, with envelope glycoprotein (Env) protruding from the surface and a conical core encapsulating the viral genome and associated proteins [49,50]. The Env trimers on the virion surface exist in various conformational states, with structural heterogeneity and a flexible stalk that allows for variable exposure of neutralizing epitopes, which may contribute to immune-evasion strategies [51]. They have also shed light on the assembly and maturation processes of HIV, particularly how the Gag polyprotein is cleaved during maturation, leading to the Env trimer gaining the motility needed to mediate membrane fusion. Additionally, cryo-ET has provided insights into HIV fusion with host cell membranes by using giant plasma membrane vesicles (or blebs) to visualize fusion intermediates [52]. Host restriction factors, such as Serinc3 and Serinc5, affect the progression of fusion, influencing the prevalence of later fusion products. Cryo-ET has also resolved the structure of the HIV-1 intasome, a crucial complex for viral integration, providing high-resolution insights into how integrase self-associates to form a functional complex [53]. This has helped to resolve previous conflicting models of intasome assembly. Overall, cryo-ET has advanced our understanding of HIV’s structural dynamics, offering detailed views of the virus’s assembly, maturation, and entry mechanisms, which are critical for guiding the development of therapeutic interventions. By combining the strengths of TEM and cryo-ET, a comprehensive understanding of HIV’s structural and functional characteristics can be reached, ultimately contributing to the fight against HIV/AIDS.

The study of giant viruses using both TEM and cryo-EM has provided significant insights into their unique structural and functional characteristics, enhancing our understanding of their biology and evolution. Light microscopy and TEM provided insights into the morphology and lifecycle of Pandoraviruses within Acanthamoeba castellanii host cells [54]. Light microscopy revealed that Pandoravirus particles are large enough to be seen as ovoid structures approximately 1 μm in length and 0.5 μm in diameter, enabling direct visualization of their multiplication in the host. TEM further elucidated the structural details and replication process, showing that the viral particles are encased in a unique tegument-like envelope composed of three layers and interrupted by an ostiole-like pore approximately 70 nm in diameter. The particles are internalized into Acanthamoeba cells through phagocytosis, fusing with the phagosome membrane and releasing their contents into the host cytoplasm. Once inside, Pandoraviruses induce significant changes in the host nucleus, leading to its degradation. The viral particles contain a diffuse interior with a spherical area of electron-dense material and exhibit a simultaneous synthesis of their tegument and internal compartments, a process that initiates at the ostiole-like apex and proceeds in a knitting-like fashion. On the other hand, cryo-EM has provided high-resolution, three-dimensional views of giant viruses, revealing intricate details that are difficult to capture with traditional TEM. Cryo-EM has confirmed the diverse morphologies of giant viruses, ranging from icosahedral to asymmetric forms, and has been particularly effective in revealing complex internal structures, including capsid architecture and genome packaging [55]. High-resolution cryo-EM studies have also shown detailed capsid protein arrangements and surface features, such as spikes, across various giant viruses. Additionally, cryo-EM has suggested novel assembly pathways, as observed in the Cafeteria roenbergensis virus capsid study, and has revealed characteristic internal membrane structures in some giant viruses, such as those in the giant Acanthamoeba polyphaga mimivirus [56]. Cryo-EM has also uncovered novel capsid protein networks and scaffold proteins of the Tokyo virus, which are critical in capsid assembly and size regulation [57]. However, studying giant viruses with cryo-EM presents specific challenges due to their large size, including issues such as the Ewald sphere effect, which can distort data and complicate the accurate reconstruction of large viral structures, and multiple electron scattering, which can introduce noise and artifacts, further complicating the visualization of the virus’s intricate details [58]. Despite these challenges, the findings provided by cryo-EM have been invaluable in advancing our understanding of the unique biology of giant viruses. Both TEM and cryo-EM have their methodological challenges when studying giant viruses due to their large size, but by employing specialized techniques, these two methods complement each other, offering a comprehensive view of the structure, assembly, and functional characteristics of giant viruses. Together, they contribute significantly to the growing knowledge of giant viruses and their evolutionary significance. Key findings of viral morphology by EM observation are summarized in Table 1.

## 3. New Insights into SARS-Cov-2 Particle Formation Obtained by TEM and ET

As mentioned above, TEM is an accessible electron microscopy analysis suitable for more comprehensive structural analysis within complex cellular environments. In our recent study focusing on the intracellular budding process of SARS-CoV-2, conventional TEM analysis successfully captured the moment when virus particles containing nucleocapsid cores budded on the lumen of vacuoles in infected Vero E6/TMPRSS2 cells [59], which was assumed to be in the ERGIC (Figure 4A) [30]. The diameter of SARS-CoV-2 particles was measured to be between 100 and 120 nm based on the TEM image. Since technological advances, including direct electron detection, automated image acquisition, and image processing resolution, have enabled better and more specialized 3D reconstructions in a single section from conventional EM images obtained via TEM, we next applied the virus particle-containing ultrathin section to ET. ET approaches are based on TEM imaging of a biological specimen at different tilt angles. ET can be applied to both plastic-embedded and frozen samples. For ET, the specimens are placed in a holder that can be tilted under the electron beam, and images are collected at angular intervals of 1–3°, generating a stack of 2D projections of a selected specimen area (Figure 2) [32,60,61,62]. The 3D images in Y and X rotations of a representative virus particle inside the vacuole obtained via ET showed that the characteristic roots of the spikes were arranged on the surface of the virion (Figure 4B,C). Figure 4D,E shows sequential 3D ET images in the Z-slice position from two different angles of the selected area, clarifying the high-electron-density structure inside the SARS-CoV-2 particles, which is considered to be the nucleocapsid structure of the budding virion. The original video source of the 3D images is available in [30].

SARS-CoV-2 particles in cytoplasmic vacuoles were found to be morphologically identical to those outside infected Vero cells [30], and a similar budding process of SARS-CoV-2 within the vacuoles was observed in the human subbronchial epithelial cell line Calu-3 via TEM analysis [31]. However, in the follow-up study, we found that a different budding status of SARS-CoV-2 particles was observed in the TEM micrographs, and ET analysis of the section revealed that two different morphological types of SARS-CoV-2 particles were present in virus-infected cells: “hollow” particles characterized by low-electron-density internal structures and high-electron-density external structures, and “dense” particles characterized by high-electron-density internal and external structures (Figure 5A,B). Importantly, most SARS-CoV-2 particles observed on the surface membrane of infected Calu-3 cells were dense particles (Figure 5B, right panel). The SARS-CoV-2 with high-electron-density structures (i.e., dense particles) observed in the vacuole and on the cell surface were morphologically identical in terms of spikes arranged on the virion surface and internal structures (Figure 5B, middle and right panels). In contrast, the hollow state of SARS-CoV-2 particles, whose inner and outer electron densities were low and high, respectively (Figure 5A and 5B left panel), was more evident in the sequential 3D ET images (Figure 5C,D). In support of this observation, when the electron-density ratio between the inside and outside of the virus particles was determined, the density ratio of SARS-CoV-2 particles within vacuoles was statistically different from that observed on the cell membrane surface [31].

Therefore, our TEM and ET results clearly demonstrated that two different types of virus particles, characterized by inner and outer electron density, are present in the vacuole of SARS-CoV-2-infected cells. Considering the fact that the intracellular vesicle, presumably the ERGIC, is the compartment where the SARS-CoV-2 virion is initially budded [8], one could envisage that the hollow particles represent immature forms of the virion, which in turn undergo the maturation process and are then released from the cell as an infectious virion, which was found as the dense virion via EM analysis [31]. Taken together, our study demonstrates that TEM and TEM-based ET analysis help to visually understand the maturation status of SARS-CoV-2 in infected cells. Since viral budding is a critical step in the viral life cycle, in which newly formed viruses are released from the host cell and initiate a new infection, if a new anti-SARS-CoV-2 agent is developed that potentially disrupts the viral budding stage, EM will be a powerful technology for directly validating the mechanism of action of the new antiviral agents.

## 4. EM as a Tool to Study the Morphological Integrity of Genetically Engineered Viruses

As shown, TEM and ET are unrivaled methods for visualizing the morphological status of virus particles inside cells. This is particularly useful for visually confirming the integrity and biological behavior of genetically engineered viruses. A virus-like particle (VLP) is a genetically engineered virus that lacks the viral genome that determines the pathogenicity and is non-infectious and non-replicable. Since the VLP is mainly produced by viral structural proteins and retains the antigenic properties of the virus, it has been considered as a promising application for the development of preventive vaccines [63,64,65]. In addition, VLPs have been used as a relevant model to study the virion entry and release steps in virus replication, and these properties are particularly useful for highly pathogenic viruses, such as SARS-CoV-2, that must be handled under the biosafety level 3 setting [66,67,68,69]. In the previous study, a VLP system whose S protein was fused with a HiBiT tag, a highly sensitive detection tag, was established by the transfection of plasmid DNA expressing SARS-CoV-2 structural (S, N, M, and E) proteins. This VLP harboring the HiBiT-fused S protein allowed for rapid and quantitative evaluation of SARS-CoV-2 virion production [70]. Our TEM analysis of ultra-thin sections of HiBiT VLP-producing HEK293T cells revealed that approximately 100 nm particles were confirmed [70], similar to the original SARS-CoV-2 particles [30]

Since the first report by Recaniello et al. demonstrating the production of infectious poliovirus from plasmid DNA [71], reverse genetics has been widely used to obtain recombinant viruses. Importantly, reverse genetics is a valuable approach that allows for the direct manipulation of viral genomes in vitro, enabling the characterization of viral genes in virus replication and pathogenesis and efficient vaccine development [72,73,74]. Chikungunya virus (CHIKV) is an enveloped RNA virus of the genus Alphavirus of the Togaviridae family that causes Chikungunya fever, which has been confirmed in more than 40 countries [75,76]. Using viral RNA from clinically isolated CHIKV strains, we generated plasmid DNA expressing a full-length CHIKV genome under the control of a polymerase II-driven cytomegalovirus promoter. Transfection of plasmid DNA encoding CHIKV genome cDNA into mammalian cells resulted in the production of infectious virus clones in the culture supernatants. In TEM analysis, the budding of 50–70 nm virus particles was clearly observed on the membrane surface of Vero cells infected with the infectious CHIKV clone [77].

A major advantage of conventional TEM is that epoxy resin-embedded samples can be stored at room temperature for long periods of time, and the same samples can be re-examined using electron microscopy. As another reverse genetics-based study, HIV-1 mutants were constructed by changing amino acids in the cleavage sites of the Gag precursor protein, and the relationship between virion maturation and genomic RNA dimerization was analyzed. HIV-1 virion morphology was shown to dramatically change during Gag processing, and based on our TEM observation of the virion core (ring-shaped, amorphous, or conical) and membrane state (thick or thin), the virion morphologies of Gag cleavage site mutants could be classified into four groups [78]. In the review article, when single low-electron-density structural images of HIV-1 particles were again obtained via TEM from the epoxy resin-embedded samples used in [78] and reconstituted by ET, the maturation status of HIV-1 virions could be observed as 3D images (the original video file of the 3D images is available in Appendix A). The 3D image of the HIV particle showed a characteristic bilayer of the viral envelope and an area of low electron density where the nucleocapsid was assumed to be located inside the virion. This retrospective morphological analysis of TEM and ET was also applied to yellow fever virus (YFV), an enveloped virus belonging to the genus Flavivirus of the Flaviviridae family. YFV is the causative agent of yellow fever, a viral hemorrhagic fever endemic to Africa and South America and characterized by high morbidity [79]. Although there is no specific antiviral drug to combat this virus, a live attenuated vaccine strain, YFV 17D, is currently widely used for the prevention of yellow fever [80]. Due to its very small size, detailed morphological data on the internal structure of YFV has been limited, even with electron microscopy studies. In the epoxy resin-embedded Vero cells infected with YFV 17D, virus particles with a diameter of approximately 50 nm were observed in the intracellular vacuole via TEM, and the internal structure of the YFV particle was confirmed in the sequential 3DET image (Appendix A).

## 5. Conclusions

It is always a natural desire of the researcher to see the virus itself. In this sense, EM played a critical role not only in the identification, but also in the molecular characterization of the causative virus, which facilitated the development of clinically available vaccines and antiviral drugs in the COVID-19 pandemic [9,25,81,82,83]. Compared to cryo-EM and cryo-EM, which are currently leading technologies in the structural biology of viruses [84,85,86,87], TEM has a drawback in that it does not visualize the virus in a manner close to its native state, as the sample must be subjected to chemical fixation, staining with contrast agents, and embedding in plastic resin. In addition, the visualization of specific protein–protein interactions and molecular architecture in the cellular environment is limited in TEM analysis. However, cryo-EM technology requires more complex and high-budget equipment and has not yet become a first-line method for diagnosis and research of the ultrastructural basis of viral infectious diseases. In addition, recent technological improvements in automated image acquisition and computer-controlled data processing have made 3D reconstructions of viruses using the TEM sample more convenient [30,31,32]. Although the need for chemical fixation of samples is a limitation via TEM analysis, this can be an advantage when the samples to be observed by EM are considered biohazardous materials, such as SARS-CoV-2 and HIV-1. In addition to fixation, plastic embedding required during TEM sample preparation, which is suitable for long-term storage of samples, is quite useful for occasional and retrospective analysis of virus-infected cells. Because of its simplicity, TEM has historically been the best choice for examining biological specimens at a subnanometer resolution, which is particularly useful in a clinical setting. Recent advances in computerized control of the EM system have enabled specialized 3D reconstructions in a single section from conventional EM images obtained by TEM. Therefore, with further development of user-assisting technologies such as automation in the sample preparation and visualization process, TEM and ET will become more of a method that links the clinical field and basic research into viral infectious diseases. Current EM technologies allow for the comprehensive analysis of not only the virus life cycle, but also the molecular details of viral proteins. A recent study showed that pharyngeal virus shedding was higher in SARS-CoV-2-infected patients than SARS patients in 2003 [88], and structural proteins such as N proteins other than S proteins have been suggested to be involved in the infectivity and pathogenicity of SARS-CoV-2 [89,90]. As mentioned at the beginning of this paper, TEM is suitable for the broader structural analysis within complex cellular systems, whereas cryo-EM is ideal for detailed structural studies at high resolution. Therefore, TEM and ET techniques will continue to play an important role in complementing the pathological and molecular biological aspects of viruses obtained by other techniques, laying the foundation for virus engineering and vaccine/drug development.

## Figures and Tables

**Figure 1 ijms-25-11762-f001:**
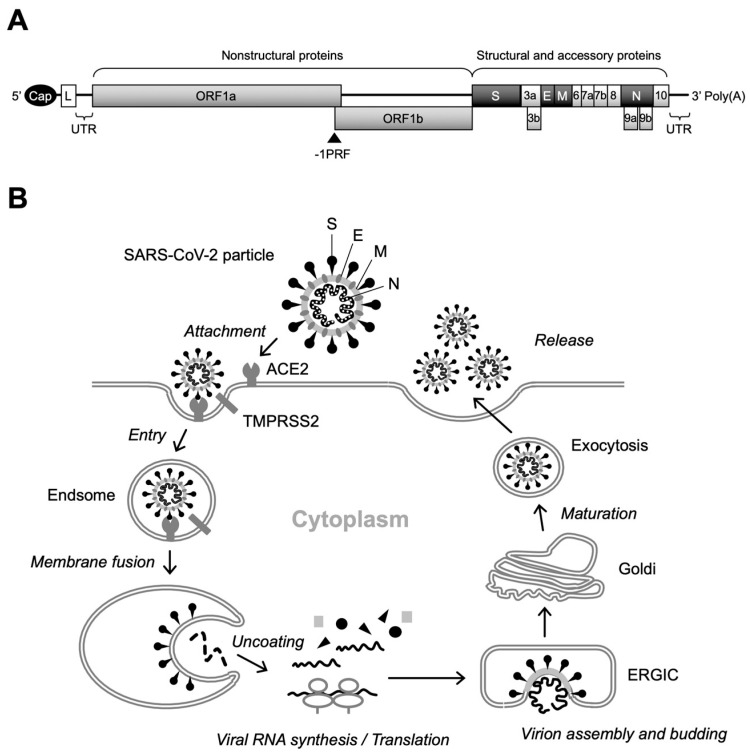
Genome structure and replication cycle of SARS-CoV-2. (**A**) Genomic organization. Sixteen NSPs are expressed as polyproteins from ORF1a and ORF1b and produced by internal cleavage by viral proteases, NSp3 (PL^pro^) and Nsp5 (M^pro^), while four structural proteins (S, E, M, and N) are expressed from individual ORFs [9]. Interspersed ORFs in the 3’ portion of the genome encode accessory proteins (ORF3a, ORF3b, ORF6, ORF7a, ORF7b, ORF8, ORF9b, ORF9c, and ORF10), which have been reported to play key roles in viral pathogenesis [10]. There is a slippery sequence (programmed -1 ribosomal frameshifting [-1PRF]) site at the overlap of ORF1a and ORF3b, where a portion of the translating ribosomes slips back one nucleotide, resulting in the polyprotein synthesis of ORF1b [11]. Cap, 5’ cap structure; L, leader sequence; UTR, untranslated region. (**B**) Overview of SARS-CoV-2 replication. After binding of the S protein to cell-surface ACE2, the attached virion is internalized into the cell via endocytosis. Specific cleavage of the S protein by cellular proteases, including TMPRSS2, then triggers fusion between the viral and endosomal membranes. Following the uncoating of the incoming virion, polyproteins of ORF1a and ORF1b are translated from the released viral RNA, which in turn is cleaved to produce NSPs that form the RTC. The viral RNA amplification process occurs on the ER membrane and translated structural proteins translocate to the lumen of the ERGIC to assemble virus particles. Finally, infectious virions are released from infected cells by exocytosis.

**Figure 2 ijms-25-11762-f002:**
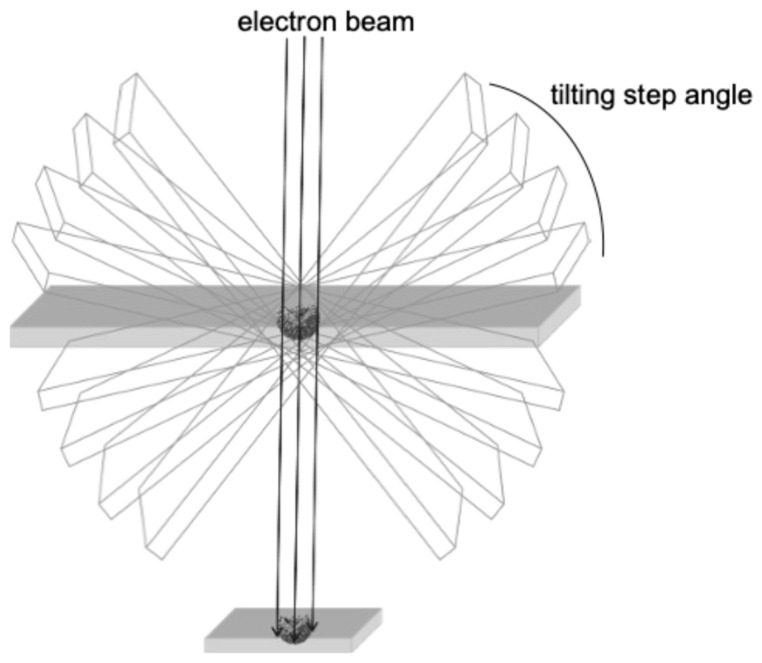
The general principle of ET.

**Figure 3 ijms-25-11762-f003:**
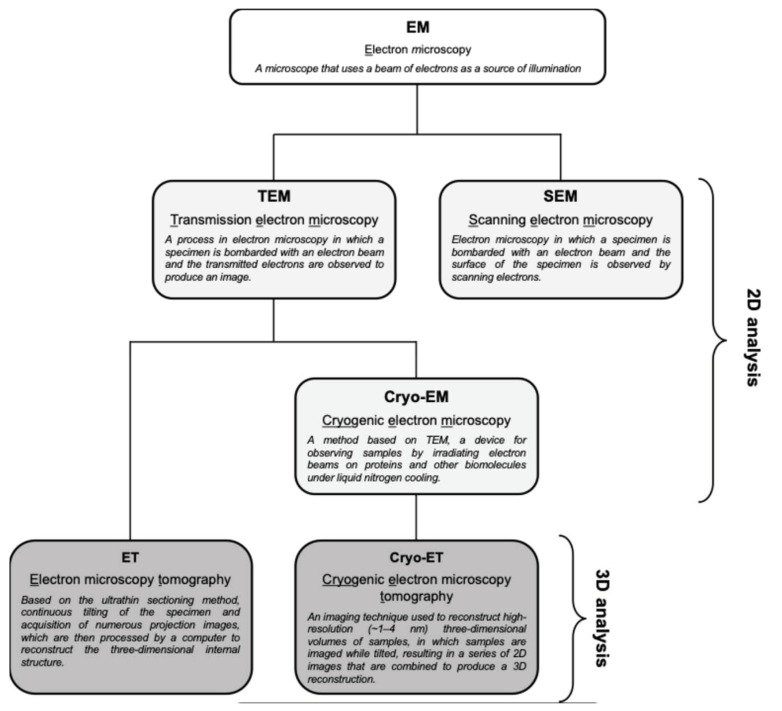
Types of EM techniques and their abbreviations. TEM provides high-resolution 2D images of ultrathin sections, which is useful for observing broader and more complex structures within a sample. ET is a TEM-based method that generates 3D reconstructions of a given area by tilting the sample and capturing multiple images. On the other hand, cryo-EM was developed for more detailed structural analysis, and cryo-ET is a specialized application of cryo-EM that reconstitutes cryo-EM data into a 3D structure. SEM is one of the electron microscopy techniques that is beyond the scope of this review.

**Figure 4 ijms-25-11762-f004:**
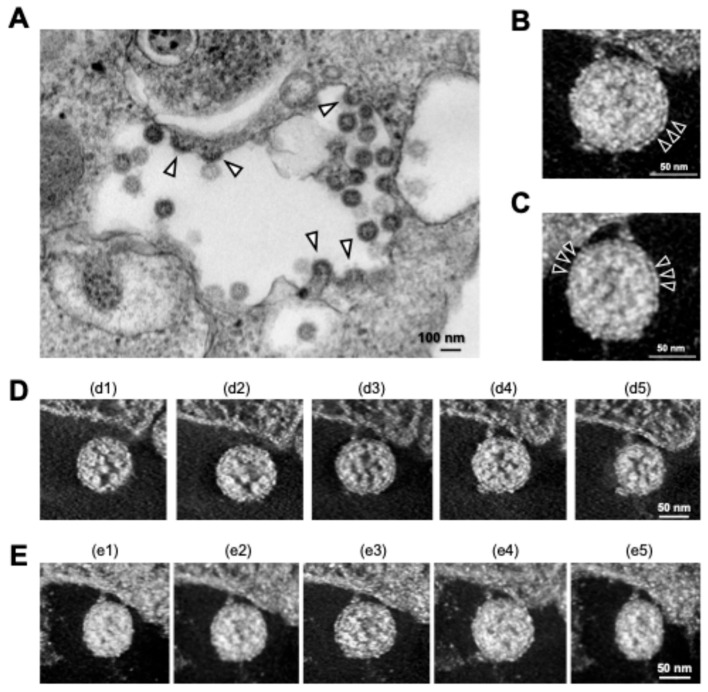
Intracellular SARS-CoV-2 image obtained via TEM and ET. (**A**) Conventional TEM image of budding virus particles (arrowheads); (**B**,**C**) 3D ET images of budding virion in Y (**B**) and X (**C**) rotation. Arrowheads indicate S proteins on the surfaces of the virions. (**D**,**E**) Sequential 3D ET images in the Z-slice position obtained from two videos (d1–d5 and e1–e5). All panels are adapted with permission from [30]. Copyright 2022 Wu et al. https://creativecommons.org/licenses/by/4.0/ (accessed on 28 October 2024).

**Figure 5 ijms-25-11762-f005:**
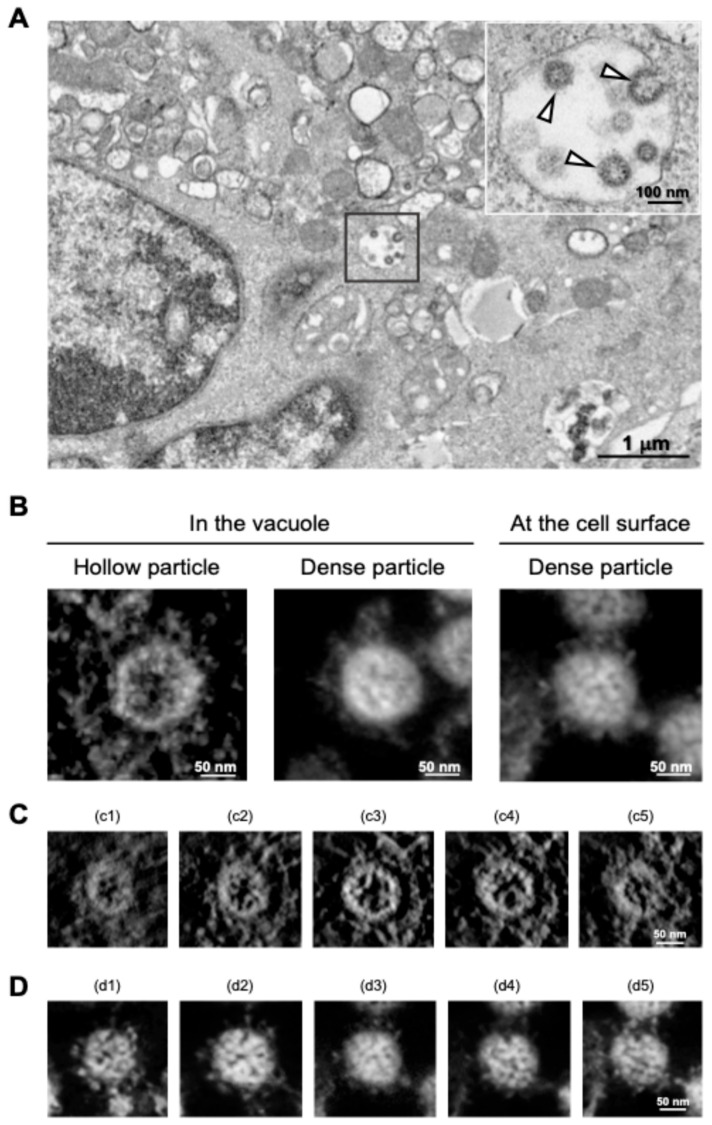
Distinct morphologies of SARS-CoV-2 observed in the vacuoles. (**A**) TEM image of hollow viral particles (white arrowheads in insert) budding within the intracellular vacuole observed near the nucleus. (**B**) Representative 3D ET images (in Y rotation) of SARS-CoV-2 particles inside the intracellular vacuole (left and middle panels) and on the surface membrane (right panel) of infected Calu-3 cells. (**C**,**D**) Z-slice position image (i.e., from c1 to c5 and d1 to d5) of hollow (**C**) and dense (**D**) particles budding within the vacuole. All panels are adapted with permission from [31]. Copyright 2024 Wu et al. https://creativecommons.org/licenses/by/4.0/ (accessed on 28 October 2024).

**Table 1 ijms-25-11762-t001:** Summary of the application of EM to virus research.

Virus	EM Methods	Findings	References
SARS-CoV-2	TEM	SARS-CoV-2 virions are spherical or pleomorphic, with diameters ranging from 80 to 120 nm.	[37]
	TEM	The internal structure of SARS-CoV-2, including the helical nucleocapsid and the replication of the virus within the cytoplasm.	[18,39]
	TEM	The exocytosis of virus particles and ultrastructural changes in infected cells.	[41]
	Cryo-ET	The roughly spherical shape of the virus, with a diameter of approximately 89.8 nm, and the conformational states of the spike proteins.	[16]
	Cryo-ET	How the virus packs its 30 kb-long RNA genome.	[26,43]
	Cryo-ET	Cryo-ET showed the formation of DMVs within infected cells and the assembly and budding of virions at the ERGIC.	[29]
HIV	TEM	HIV particles are spherical, possess an envelope derived from the host cell membrane, and contain a cone-shaped core encapsulating the viral RNA genome.	[44]
	TEM	Mature HIV-1 particles contain two copies of RNA strands within the core, forming an interwound, coiling structure.	[45]
	TEM	The pleomorphic nature of HIV capsids.	[46]
	TEM	Inositol hexakisphosphate (IP6) stabilized the capsid structure, preventing premature disassembly during sample preparation.	[47,48]
	Cryo-ET	High-resolution 3D views of HIV, revealing the Env glycoproteins on the virion surface and a conical core encapsulating the viral genome.	[49,50]
	Cryo-ET	The structural heterogeneity of Env trimers, with a flexible stalk that allows for variable exposure of neutralizing epitopes, contributing to immune-evasion strategies.	[51]
	Cryo-ET	Fusion intermediates were visualized by using giant plasma membrane vesicles. Serinc3 and Serinc5 affect the progression of fusion at multiple steps.	[52]
	Cryo-ET	The structure of the HIV-1 intasome, providing high-resolution insights into how integrase self-associates to form a functional complex.	[53]
Giant Viruses	Light microscopy and TEM	The morphology and lifecycle of Pandoraviruses within host cells, showing viral particles 1 µm in length and 0.5 µm in diameter. The viral particles are encased in a unique tegument-like envelope and their entry into host cells occurs through phagocytosis.	[54]
	Cryo-EM	Novel capsid protein networks and scaffold proteins of the Tokyo virus, which are critical in capsid assembly and size regulation.	[57]
	Cryo-EM	The capsid structure of the Cafeteria roenbergensis virus and suggested novel assembly pathway for giant viruses.	[56]

## Data Availability

All data are included in this article and Appendix A.

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
