# Peer review of "Electron Tomography as a Tool to Study SARS-CoV-2 Morphology"

_ijms, 2024, doi:10.3390/ijms252111762_

Round 1
Reviewer 1 Report
Comments and Suggestions for Authors
The review article submitted by Wu et al entitled “Electron tomography as a tool to study SARS-CoV-2 morphology” describes the benefits of Electron microscopy in the field of virology as well as their recent findings on morphologically distinct types of SARS-CoV-2 particles using TEM combined with electron tomography (ET). The review article is very well
written with good new information. The authors have detailed on both SARS CoV-2 as well as about EM in the beginning and then moved to the specific roles of EM. I would encourage the authors to introduce a table with details of what type of study done with various EM across the world in terms of medically significant viral infection, so as it will be easy for the readers for finding the references.
Author Response
The review article submitted by Wu et al entitled “Electron tomography as a tool to study SARS-CoV-2 morphology” describes the benefits of Electron microscopy in the field of virology as well as their recent findings on morphologically distinct types of SARS-CoV-2 particles using TEM combined with electron tomography (ET). The review article is very well written with good new information. The authors have detailed on both SARS CoV-2 as well as about EM in the beginning and then moved to the specific roles of EM.
We appreciate the reviewer’s positive evaluation and helpful comments. Italics refer to new locations in the revised manuscript.
Point 1: I would encourage the authors to introduce a table with details of what type of study done with various EM across the world in terms of medically significant viral infection, so as it will be easy for the readers for finding the references
Response 1: We fully agree with the reviewer’s suggestion. We have added a table in the revised manuscript summarizing what types of EM techniques have been used to study medically important viruses (page 8 “Table 1. Summary of the application of EM to virus research”).
Reviewer 2 Report
Comments and Suggestions for Authors
Reviewer report on Manuscript ‘Electron tomography as a tool to study SARS-CoV-2 morphology’
In this Review paper authors focus our attention on the usefulness of conventional transmission electron microscopy in the analysis of the viral ultrastructure. In addition, authors have shown that transmission electron microscopy can also be applied to 3D morphological analysis of SARS-CoV-2 particles (electron tomography [ET]) and revealed the presence of different types of virus particles in infected cells, which would indicate the different maturation states of virus particles within cells. Therefore, TEM and its 3D application to ET still serve as a complementary method for in situ analysis of the behavior of SARS-CoV-2.
This article is well-designed, well-illustrated and is very interesting, from the point of view of bioanalytical chemistry. The research is in the scope of the journal. Therefore, the manuscript eventually can be published after some additional minor corrections and improvements:
Introduction of the manuscript could be advanced by overview of reviews on application of alternative methods applied for the determination of SARS-CoV-2 virus (Biosensors for the Determination of SARS-CoV-2 Virus and Diagnosis of COVID-19 Infection. International Journal of Molecular Sciences 2022, 23, 666.).
Some more additional figures could be added to the manuscript.
Conclusions could be advanced by additional insights for future developments related to the application of TEM in research of viruses and viral infections.
Author Response
Reviewer report on Manuscript ‘Electron tomography as a tool to study SARS-CoV-2 morphology’
In this Review paper authors focus our attention on the usefulness of conventional transmission electron microscopy in the analysis of the viral ultrastructure. In addition, authors have shown that transmission electron microscopy can also be applied to 3D morphological analysis of SARS-CoV-2 particles (electron tomography [ET]) and revealed the presence of different types of virus particles in infected cells, which would indicate the different maturation states of virus particles within cells. Therefore, TEM and its 3D application to ET still serve as a complementary method for in situ analysis of the behavior of SARS-CoV-2.
This article is well-designed, well-illustrated and is very interesting, from the point of view of bioanalytical chemistry. The research is in the scope of the journal. Therefore, the manuscript eventually can be published after some additional minor corrections and improvements:
We appreciate the reviewer’s positive assessment of this manuscript. Italics refer to new locations in the revised manuscript.
Point 1: Introduction of the manuscript could be advanced by overview of reviews on application of alternative methods applied for the determination of SARS-CoV-2 virus (Biosensors for the Determination of SARS-CoV-2 Virus and Diagnosis of COVID-19 Infection. International Journal of Molecular Sciences 2022, 23, 666.).
Response 1: In accordance with the reviewer’s suggestion, we have added sentences in the Introduction (page 3, lines 120-122) with the suggested reference (reference 20).
Point 2: Some more additional figures could be added to the manuscript.
Response 2: To facilitate the reader’s understanding, we have added two figures in the revised manuscript (Figures 3 and 5). The new Figure 3 shows brief explanations of the electron microscopy techniques focused on in this review and their classification (page 5). In the new Figure 5 (page 10), we added a new panel of a conventional TEM image showing the “hollow” SARS-CoV-2 particles (characterized by low-electron-density internal structures and high-electron-density external structures) observed in intracellular vacuoles, in the original Figure 4.
Reviewer 3 Report
Comments and Suggestions for Authors
This is a good and interesting review paper but there are a few shortcomings that need to be addressed.
1) In line 30, it mentions that COVID-19 pandemic is ongoing. This is wrong. WHO declared COVID-19 pandemic has ended in May 2023 but many experts believe that COVID-19 is endemic, which means that it will probably be entrenched in the world permanently like influenza.
https://www.ncbi.nlm.nih.gov/pmc/articles/PMC9361413/
2) This paper is very interesting to people who are involved in virus research but do not have much background in microscopy. But I find it hard to read the paper because of the terminology. I recommend that the authors have a table containing the abbreviation for the terminology (e.g cry-EM, EM, TEM, ET,..etc) on one column. Another column will be the full name of the terminology and a third column with a brief description of each.
3) The paper makes an interesting point that modern microscopy is able to observe the life-cycle of the virus. This is important as Wolfel et al has shown that COVID-19 patients sheds much more particles than the 2003 SARS patients. The cause of the greater infectiousness/virulence is likely to lie in the life-cycle of the virus. But the life-cycle does not just involve S and viral entry. Several papers point to M and N as the primary cause of infectiousness and virulence.
https://virologyj.biomedcentral.com/articles/10.1186/s12985-023-01968-6
https://journals.plos.org/plospathogens/article?id=10.1371%2Fjournal.ppat.1010627&fbclid=IwY2xjawF2b-ZleHRuA2FlbQIxMAABHbv8H7-819wnt2a7XQgVx-jFCorj64Q1hO582d9KUccf22s5I17ydftUIQ_aem_IkQYe8G1xB0cIXG6fdhRlg
https://pubmed.ncbi.nlm.nih.gov/39062780/
https://pubmed.ncbi.nlm.nih.gov/32235945/
https://www.ncbi.nlm.nih.gov/pmc/articles/PMC4147684/
4) Related to (3), while the paper mentions about virus structure, very little has been mentioned about the structures of the viral proteins. I am aware that modern EM is so powerful that it can determine the structure of a protein to a high resolution. Can EM determine the structures of N at various stages of the virus life-cycle within a cell? This is important as there are papers showing that the structures or lack of structure of N/M is the determinant of infectivity. Protein structure is therefore important.
https://www.nature.com/articles/s41586-020-2833-4
Author Response
This is a good and interesting review paper but there are a few shortcomings that need to be addressed.
We thank the reviewer for the positive and helpful comments. Italics refer to new locations in the revised manuscript.
Point 1: In line 30, it mentions that COVID-19 pandemic is ongoing. This is wrong. WHO declared COVID-19 pandemic has ended in May 2023 but many experts believe that COVID-19 is endemic, which means that it will probably be entrenched in the world permanently like influenza.
https://www.ncbi.nlm.nih.gov/pmc/articles/PMC9361413/
Response 1: We apologies for the inaccurate descriptions of the current COVID-19 situation. In response to the reviewer’s comment, we have changed the first paragraph of the Introduction (page 1, lines 27–42).
Point 2: This paper is very interesting to people who are involved in virus research but do not have much background in microscopy. But I find it hard to read the paper because of the terminology. I recommend that the authors have a table containing the abbreviation for the terminology (e.g cry-EM, EM, TEM, ET,..etc) on one column. Another column will be the full name of the terminology and a third column with a brief description of each.
Response 2: We agree with the comment that the explanations of the electron microscopy (EM) methods and their abbreviations will be helpful to the readers. In the revised version of the manuscript, we have included the summary as Figure 3 (page 5), especially so that their classification can be easily recognized.
Points 3 and 4: The paper makes an interesting point that modern microscopy is able to observe the life-cycle of the virus. This is important as Wolfel et al has shown that COVID-19 patients sheds much more particles than the 2003 SARS patients. The cause of the greater infectiousness/virulence is likely to lie in the life-cycle of the virus. But the life-cycle does not just involve S and viral entry. Several papers point to M and N as the primary cause of infectiousness and virulence.
https://virologyj.biomedcentral.com/articles/10.1186/s12985-023-01968-6
https://journals.plos.org/plospathogens/article?id=10.1371%2Fjournal.ppat.1010627&fbclid=IwY2xjawF2b-ZleHRuA2FlbQIxMAABHbv8H7-819wnt2a7XQgVx-jFCorj64Q1hO582d9KUccf22s5I17ydftUIQ_aem_IkQYe8G1xB0cIXG6fdhRlg
https://pubmed.ncbi.nlm.nih.gov/39062780/
https://pubmed.ncbi.nlm.nih.gov/32235945/
https://www.ncbi.nlm.nih.gov/pmc/articles/PMC4147684/
Related to (3), while the paper mentions about virus structure, very little has been mentioned about the structures of the viral proteins. I am aware that modern EM is so powerful that it can determine the structure of a protein to a high resolution. Can EM determine the structures of N at various stages of the virus life-cycle within a cell? This is important as there are papers showing that the structures or lack of structure of N/M is the determinant of infectivity. Protein structure is therefore important.
https://www.nature.com/articles/s41586-020-2833-4
Responses 3 and 4: We appreciate the very insightful comments, which will certainly improve our manuscript. We agree that the structure of not only the S protein but also the M and N proteins play an important role in the infectivity and pathogenicity of SARS-CoV-2, and it would be useful if the structural changes of these proteins could be followed by electron microscopy. In this respect, we believe that cryo-EM, which is ideal for detailed structural analysis, could complement the data obtained by TEM analysis, which is suitable for broader structural analysis within cells. Therefore, in accordance with the reviewer’s comments, we have added new sentences and references in the revised manuscript (from page 12, line 572 to page 13, line 582).
Reviewer 4 Report
Comments and Suggestions for Authors
The authors summarized the utility of electron tomography (ET) to visualize SARS-CoV2 virions in their relatively native state as compared to the electron microscopy (EM). While the review is interesting as well as useful, the manuscript is well written. This reviewer has only few minor comments:
1. Abstract can be improved by incorporating the conclusions of this study.
2. Conclusions can be improved. The initial sentences read like introduction.
Author Response
The authors summarized the utility of electron tomography (ET) to visualize SARS-CoV2 virions in their relatively native state as compared to the electron microscopy (EM). While the review is interesting as well as useful, the manuscript is well written. This reviewer has only few minor comments:
We greatly appreciate the positive evaluation by the reviewer. Italics refer to new locations in the revised manuscript.
Point 1: Abstract can be improved by incorporating the conclusions of this study.
Response 1: The last part of the abstract was changed by incorporating the conclusion of this review (page 1, lines 17–22).
Point 2: Conclusions can be improved. The initial sentences read like introduction.
Response 2: According to the reviewer’s suggestion, the initial sentences of the Conclusion in the original manuscript were removed. In addition, the Conclusion section was largely improved by including the suggestions/comments raised by other reviewers (page 12, lines 547–550; page 12, line 565–page 13, line 582).
Round 2
Reviewer 3 Report
Comments and Suggestions for Authors
Improvements seen